# Dexmedetomidine Effectively Sedates Asian Elephants (*Elephas maximus*)

**DOI:** 10.3390/ani12202787

**Published:** 2022-10-15

**Authors:** Nithidol Buranapim, Pawinee Kulnanan, Kullapassorn Chingpathomkul, Taweepoke Angkawanish, Saran Chansitthiwet, Warangkhana Langkaphin, Petthisak Sombutputorn, Natcha Monchaivanakit, Kankawee Kasemjai, Kittikul Namwongprom, Khajohnpat Boonprasert, Pakkanut Bansiddhi, Niyada Thitaram, Patrick Sharp, Cholawat Pacharinsak, Chatchote Thitaram

**Affiliations:** 1Department of Companion Animal and Wildlife Clinic, Faculty of Veterinary Medicine, Chiang Mai University, Chiang Mai 50100, Thailand; 2Center of Elephant and Wildlife Health, Chiang Mai University, Chiang Mai 50100, Thailand; 3Elephant Hospital, National Elephant Institute, Forest Industry Organization, Lampang 52190, Thailand; 4Office of Research and Economic Development, University of California, Merced, CA 95343, USA; 5Department of Comparative Medicine, Stanford University, Stanford, CA 94305, USA

**Keywords:** Asian elephants, dexmedetomidine, sedation

## Abstract

**Simple Summary:**

Sedation in standing procedures is commonly performed in elephants. To successfully and safely initiate standing in elephants, chemical restraint drugs should provide sufficient sedation with minimal complications (i.e., recumbency, significant physiologic alterations, and prolonged recovery). This study investigated the sedative effects of dexmedetomidine in Asian elephants. Results suggest dexmedetomidine 1–2 µg/kg provides effective sedation. However, we suggest a single intramuscular dexmedetomidine injection of 2 µg/kg for approximately 70 min of sedation. This is the first study to demonstrate dexmedetomidine use in Asian elephants.

**Abstract:**

This study investigated the sedative effects of dexmedetomidine in Asian elephants. We hypothesized that 2 µg/kg dexmedetomidine would provide sufficient standing sedation. A crossover design study was performed in three Asian elephants. Each elephant was assigned to 1 of 3 treatment groups—1 (D1), 1.5 (D1.5) or 2 (D2) µg/kg dexmedetomidine (intramuscular injection, IM) with a two-week ‘washout period’ between doses. Elephants were monitored for 120 min. At 120 min (Ta), atipamezole was administered IM. Sedation and responsiveness scores were evaluated. Physiological parameters (pulse rate, respiratory rate, and %SpO_2_) and clinical observations were monitored during the study and for 3 days post drug administration. D2 provided the longest sedation (approximately 70 min), compared to D1 and D1.5. After Ta, each elephant’s sedative stage lessened within 10–15 min without complications. No significant abnormal clinical observations were noted throughout and during the 3-days post study period. These data suggest that a single 2 µg/kg IM dexmedetomidine injection provides sufficient standing sedation for approximately 70 min in Asian elephants.

## 1. Introduction

Due to the large size of elephants, and to minimize general anesthesia-related complications, standing procedures are commonly performed. To successfully perform standing procedures, restraint is crucial. Correct restraint methods should provide safety to both humans and animals. However, it is challenging to execute elephant restraint safely. Chemical immobilization is one of the most common restraint methods. Ideal chemical restraint drugs should provide sufficient standing sedation with minimal complications (i.e., recumbency, significant physiologic alterations, and prolonged recovery). To chemically immobilize elephants, α2 adrenergic agonists, i.e., xylazine [1], detomidine [2], and medetomidine [3,4] have been used. Given the efficacy of these α2 adrenergic agonists in elephants, there have been periodic market shortages of these drugs; therefore, these compounds may be unavailable in some countries, thereby necessitating an alternative.

Dexmedetomidine, an α2 adrenergic agonist, has been used widely in domestic and exotic animals including wildlife, dogs [5,6], cats [7], rodents [8,9], dairy and dromedary calves [10,11], horses [12], miniature donkeys [13], and captive tigers [14]. Like other α2 adrenergic agonists, it provides sedation, muscle relaxation, and analgesia [15]; there- fore, it has gained popularity over the last 10 years. Dexmedetomidine is a sympatholytic drug specific to α2 receptors. Its α2:α1 ratio selectivity is 1620:1, which differs from xylazine (160:1), clonidine (220:1), detomidine (260:1), and medetomidine (1620:1) [15]. Due to its high α2 receptor specificity compared to other α2 adrenergic agonists, at clinical practice doses, dexmedetomidine is considered a potent α2 adrenergic agonist requiring a lower dose and volume to achieve a similar sedation level. In addition, dexmedetomidine’s high selectivity should be beneficial due to providing sufficient sedation with minimal side effects and complications. However, the effects of dexmedetomidine in elephants is unknown. 

The aim of this study was to investigate the sedative effects of dexmedetomidine in elephants. We hypothesized that 2 µg/kg dexmedetomidine would provide moderately heavy sedation for a longer duration than 1 or 1.5 µg/kg dexmedetomidine in Asian elephants. To our knowledge, this is the first report evaluating dexmedetomidine’s sedative effects in Asian elephants.

## 2. Materials and Methods

### 2.1. Animals

Three healthy, non-musth male Asian elephants (elephant 1 body weight (BW) 2870 kg, age 28 years; elephant 2 BW 2900 kg, age 20 years; elephant 3 BW 3865 kg, age 27 years, average age, 25.7 ± 5.1 years old, average BW 2800 ± 900.5 kg) from the National Elephant Institute, Forest Industry Organization, Lampang, Thailand, were used in this study. All animal procedures were approved by the Animal Ethics Committee, Faculty of Veterinary Medicine, Chiang Mai University, Thailand (S23/2562). All elephants served as tourist trekking animals or performed in shows for no more than 3 h each day (between 08.00 and 15.00 h). During the day, and when not working, bulls were maintained in a shed among different male groups and provided grass, bananas, sugar cane, hay, rice grains, and water ad libitum. Later on in the afternoon, they were separately tethered with a 5 m long chain near a caretaker’s (mahout) house overnight. The animals received a semiannual physical examination conducted by a veterinarian. Blood from each bull was collected at the National Elephant Institute, where hematological and biochemical profiles were performed. All elephants used in this study were deemed healthy. On the day of the study, animals were kept fasted (food, 8–12 h; water, 1–3 h) and weighed beforehand. Fifteen minutes before drug administration, elephants were brought into a stall; an intravenous catheter was used (20G, left or right auricular vein) which administered normal saline (0.9% NaCl), 5 mL/kg/hr.

### 2.2. Study Design

Elephants were randomly assigned to 1 of the following 3 treatment groups: (1) dexmedetomidine 1 µg/kg (D1, dexmedetomidine HCL, 0.5 mg/mL, Dexdomitor^®^; Orion Corporation, Espoo, Finland); (2) dexmedetomidine 1.5 µg/kg (D1.5); or (3) dexmedetomidine 2 µg/kg (D2). Dexmedetomidine was injected (T0) intramuscularly (IM, 18G, 1.5”) into the triceps brachii muscle. All elephants were supported with chest belts hanging from a pulley to avoid an accidental fall. This study employed a crossover experimental design, with each drug dose administered according to a 14-day washout interval; thus, sedation was conducted three times in each bull (14 days apart) as outlined in Table 1.

At the end of each experiment (120 min, Ta), atipamezole (5 mg/mL, Antisedan^®^; Orion Corporation, Espoo, Finland), was administered (ten times the dexmedetomidine dosage used, 18G, 1.5”) into the ipsilateral triceps brachii muscle, but at different injection sites. Elephants recovered in the sedation stall. Once fully recovered, they were returned to their stalls and monitored daily for 3 consecutive days.

Throughout the study, elephants were monitored for sedation and vital sign parameters assessment at pre- (−15, −10, and −5 min) and post-dexmedetomidine injection (T0) every 5 min for 120 min. A responsiveness assessment was performed at pre- and post-drug administration every 15 min until min 120; then, an atipamezole was administered (Ta). They were also monitored post atipamezole administration for 30 min, as illustrated in Figure 1.

### 2.3. Sedation Assessment

Sedation assessment criteria [6 criteria; from 0 (not sedated) to- 18 (most sedated)] are shown in Table 2. Four sedation levels were classified as follows: (1) slight (scores 0–5), (2) mild (scores 6–11), (3) moderate (scores 12–15), and (4) heavy sedation (score 16–18). Sedation assessments were performed every 5 min from −15 to 150 min [−15, −10, −5 min prior to drug administration (T0) and every 5 min for 150 min] as shown in Figure 1.

### 2.4. Responsiveness Assessment

The criteria [6 criteria; from 0 (not responsive) - 18 (responsive)] are shown in Table 3. A responsive assessment was performed twice prior to drug administration (T0) and every 15 min for 150 min as shown in Figure 1. Three responsiveness levels were classified as follows: (1) slight (scores 0–6), (2) moderate (scores 7–11), (3) strong (scores 12–18).

Three blinded, experienced elephant veterinarians evaluated sedation and responsive assessment.

### 2.5. Vital Sign Parameters Assessment

Vital sign parameters [Pulse rate (PR), respiratory rate (RR), body temperature, and capillary refill time (CRT)] were assessed every 5 min from −15 to 150 min [−15, −10, −5 min prior to drug administration (T0) and every 5 min for 150 min], as presented in Figure 1.

### 2.6. Clinical Observation

Elephants were monitored for other clinical signs (hypersalivation, shivering, vomiting, nystagmus, and urination) throughout the study and at 3 days post-injection. 

### 2.7. Data Analysis

Data are presented as mean ± standard error of the mean (SEM). The differences in the means of sedation score and vital sign values among drug dose groups were compared using ANOVA. Since the Latin square design is employed, the ANOVA model corresponds to the design as follows:$$y_{ijk}=\mu +\beta_{i}+\gamma_{j}+\tau_{k}+\varepsilon_{ijk}$$
where $y_{ijk}$ is the observation from i animal receiving drug k at period j, while μ is the overall mean. The term $\beta_{i}$ denotes the individual effect of i animal (i = 1, 2 and 3) and $\gamma_{j}$ represent effects from period (j = 1, 2 and 3), while τ is the effect from drug doses (i = 1 µg/kg, 1.5 µg/kg and 2 µg/kg). The term $\varepsilon_{ijk}$ is residual and it was assumed that $\varepsilon ∼NID\left(0,\sigma^{2}\right)$. For multiple comparisons (Post Hoc), the Bonferroni method was used. The differences in means among stages relevant to the drug administration stages (1 = before, 2 = during and 3 = after) across all doses were compared. The level of significance was set as α=0.05 for all statistical analyses.

## 3. Results

Elephants’ weights were not significantly different from week 1 to week 6 throughout the study.

### 3.1. Sedation Score

Sedation scores (scores 0–5) were observed in all three groups immediately after dexmedetomidine administration and lasted for 10 ± 1.0 (D1), 10 ± 0.1 (D1.5), and 10 ± 0.2 (D2) min. A mild sedation (scores 6–11) duration among the three groups was also observed for 10 ± 0.1 (D1), 10 ± 0.9 (D1.5), and 5 ± 0.4 (D2) min. The onset of moderate-heavy sedation (scores 12–18) was observed for 20 ± 1.5 (D1), 20 ± 2.4 (D1.5), 15 ± 0.6 (D2) min. The duration of moderate-heavy sedation was longer in D2 (70 ± 2.5 min) vs. D1 (50 ± 2.0 min) and D1.5 (55 ± 1.9 min).

To lighten the sedative plane, atipamezole was administered. After atipamezole administration (Ta), there was no difference between elephants regarding the onset of slight sedation (lightened sedation plane towards recovery) in the three groups [10 ± 0.5 (D1), 15 ± 0.8 (D1.5), 15 ± 0.1 (D2) min], as shown in Figure 2.

### 3.2. Responsiveness Score

A lower responsiveness score (slight responsive) correlates with heavier sedation. The duration of elephants presenting with a continuously low responsiveness score [0–6 (slightly responsive)] was longer in D2 (60 ± 3.0 min) vs. the D1.5 (45 ± 1.1 min) and D1 (15 ± 2.2 min) elephants. D1 elephants returned to a responsiveness score of 12–18 (strong responsive, lightened sedation plane) as early as 90 min after a dexmedetomidine injection. Immediately after Ta at 120 min, the D1.5 and D2 elephants returned to a responsive score of 12–18 (lightened sedation plane), as shown in Figure 3.

### 3.3. Vital Sign

At T0, the pulse rates among the three groups were 37 ± 0.4 (D1), 30 ± 1.2 (D1.5), and 25 ± 2.2 (D2) bpm. The respiratory rate of each group was 7 ± 0.1 (D1), 8 ± 1.0 (D1.5), and 6 ± 2.3 (D2) tpm. The pulse rate and the respiratory rate in three groups were measured throughout the study as presented in Figure 4.

The %SpO_2_ of all three groups exceeded 95% throughout the study (data not shown).

### 3.4. Clinical Observation

Clinical observation: Hypersalivation and periodic snoring were observed in all elephants at all doses, and they were not noticeable by the end of the study. There were no other abnormal clinical signs (shivering, vomiting, or nystagmus) observed throughout the study or during the 3 day post-study period.

## 4. Discussion

This study demonstrated that 2 µg/kg dexmedetomidine successfully provided moderate to heavy sedation for 70 ± 2.5 min, which was longer than for the 1 µg/kg (50 ± 2.0 min) or 1.5 µg/kg (55 ± 1.9 min) doses in Asian elephants. There was no difference in recovery time (return to slight sedative stage) among the three groups. There was no difference in pulse rate, respiratory rate, and %SpO_2_ among the three groups. There were no significant abnormal clinical observations during the study and post-study period. The current data support our hypothesis that 2 µg/kg dexmedetomidine provides moderate to heavy sedation for a longer period than 1 or 1.5 µg/kg doses in Asian elephants.

To our knowledge, this is the first study to examine the effects of dexmedetomidine use in Asian elephants. The aim of this study was to investigate the effectiveness of a relatively new sedative drug, dexmedetomidine, using three different dosages in elephants. This study included modified criteria for sedative scores, responsiveness scores, and vital sign parameters (pulse rate, respiratory rate and %SpO_2_). Due to dexmedetomidine’s high α2 receptor selectivity, it was expected to provide sufficient sedation with minimal side effects. In this study, D2 provided the longest moderate–heavy sedation (70 min) without significant abnormal clinical observations. Only mild hypersalivation and snoring were observed in the three groups and these effects subsided by the end of the study and did not reoccur in the 3-day post-procedure period.

Because ideal chemical restraint drugs should provide sufficient sedation with minimal side effects, choosing the doses and the route of administration requires strong consideration of potential side effects and practicality. Most sedatives provide sedation in a dose-dependent manner; therefore, a higher dose should elicit heavier sedation, and possibly, a higher incidence of side effects. Because the actual dexmedetomidine dose in Asian elephants is unknown, we chose doses based on the closely related α_2_ adrenergic agonist, medetomidine. Typically, medetomidine intramuscular (IM) elephant’s doses are 2–4 µg/kg [4]; therefore, dexmedetomidine doses are 1/3–1/2 of the medetomidine dose. The three doses studied spanned the range of 1, 1.5, and 2 µg/kg to safely elucidate the lowest possible dose yielding moderate–heavy sedation. To perform standing procedures in elephants, the most practical and clinical drug administration route is IM injection into the triceps brachii muscles. Additionally, this study employed a crossover design where another dexmedetomidine dose occurred every two weeks in the three elephants. Although dexmedetomidine’s ‘washout period’ is unknown in elephants, a two-week period was chosen based on data from other species (e.g., dogs) [6,17].

When evaluating sedative drugs, it is vital to observe and assess potential drug complications. This current study included such assessments of physiological parameters (pulse rate, respiratory rate, and %SpO_2_), under clinical observation, extending into a 3-day post drug administration period. Like other α2 adrenergic agonists, dexmedetomidine can cause dose-dependent cardiorespiratory depression [18]. Previous elephant studies using α2 adrenergic agonists, such as xylazine, reported respiratory depression, bradycardia, and penile prolapse [4]. Pulse and respiratory rates were not significantly altered throughout the study. In addition, %SpO_2_ remained above 95% in room air throughout the study. Although 100% O_2_ supplementation is strongly encouraged when possible, it was not provided because we aimed to mimic the clinical conditions of the farm or field where 100% O_2_ may be unavailable. Neither apnea nor penile prolapse were observed. Hypersalivation and snoring were observed, although neither were noticeable at the study’s end or during the three-day post-procedure period. Although blood pressure was not recorded, dexmedetomidine has been reported to cause hypertension followed by hypotension [19]. Due to the study’s duration (150 min), and to avoid hypotension, IV fluid was administered. Profound sedation is an α2 adrenergic agonist side effect, especially when combined with opioids [2]. To avoid an elephant’s accidental recumbency or fall that could be harmful to the animals, a sling supported the animals throughout. The sling was used out of an abundance of caution. In other species, α2 adrenergic antagonist administration is recommended after procedures where α2 adrenergic agonists are administered [20]. Commonly used α2 adrenergic antagonists are atipamezole or yohimbine. Due to its high selectivity [α2:α1 ratio—atipamezole 8526:1 vs. yohimbine 40:1] [21], atipamezole was chosen to hasten recovery. The atipamezole dosage selected was based on 10 times the dexmedetomidine dosage administered [22]. This current study demonstrated that after Ta, the time to achieve slight sedation scores (a lightened sedative plane) was not different amongst the three groups (10–15 min). All elephants recovered uneventfully and continued to recover well, with no abnormal clinical observations detected following the -procedure throughout the three-day post-drug administration period.

To evaluate sedative effects, sedative and responsiveness scores are commonly used; therefore, we modified sedative and responsiveness scores to fit the study’s needs. Previous studies showed that these scores are reliable metrics to evaluate elephant sedative quality. Many commonly performed procedures, (e.g., physical examination, eye examination, oral examination, blood collection, or minor surgical procedures) require moderate–heavy sedation. Therefore, the moderate–heavy sedation level, together with low responsiveness scores (4–6 or lower; less responsive) was selected to achieve sufficient sedation. The D2 group provided the longest sedation with continuous low responsiveness scores (4–6 or lower). This current study confirms that these scores were useful to evaluate dexmedetomidine’s effectiveness in sedating elephants. During the experiment, we tried to avoid all stimulants, e.g., noise, human voice, movement, etc. However, the snoring sound of another elephant under sedation occurred, which stimulated the response of another elephant during the experiment. The snoring did not affect the results, as it occurred for a second and then stopped.

In general, the neuroleptanalgesia concept (a combination of sedatives and opioids) is encouraged to elicit profound sedation, muscle relaxation, and analgesia. The combination requires lower doses of sedatives and opioids, thus minimizing possible side effects. Because the study’s aim was to exclusively evaluate dexmedetomidine’s sedative effects, a neuroleptanalgesia combination was not included. Although this current study did not evaluate dexmedetomidine-induced analgesia, dexmedetomidine reportedly provides analgesia in several animal models [23,24,25,26,27,28,29].

There were several limitations to our study. First, only healthy male Asian elephants were used. Anesthetics have been reported to have different effects in males than in females [30,31]. It is possible that different sedation levels occur between sexes. Second, dexmedetomidine was administered IM in a single dose. Other administration routes (e.g., subcutaneous, intravenous) or continuous rate infusions have been used in other species [32,33,34], which can be studied further. Third, other parameters, i.e., blood pressure, electrocardiogram, or analgesia levels were not included in the study. Because α2 adrenergic agonists reportedly cause arrhythmias and biphasic blood pressure in other species [35,36], it is possible that dexmedetomidine may have caused undetected complications in the elephants in the current study. Fourth, a combination of other anesthetics/analgesics, with minimized doses and side effects was excluded. Fifth, the number of elephants was limited due to the number of elephants available to use in the experimental study. The advantage in this study was to compare sedative effects from three different doses used on the same elephant.

Dexmedetomidine is an effective α_2_ adrenergic agonist for procedures requiring moderate-heavy sedation in elephants. Its benefits are as follows: (1) continuous 70 min sedation with uneventful recovery (within 10 min) after atipamezole administration; (2) unaltered physiological parameters; and (3) insignificant clinical side effects. Further evaluation is needed to titrate the dose when combined with other anesthetics. In conclusion, our data indicate that 2 µg/kg dexmedetomidine (IM) provides an effective sedation lasting approximately 70 min in Asian elephants.

## Figures and Tables

**Figure 1 animals-12-02787-f001:**
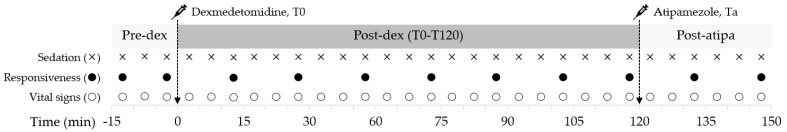
Experimental design of the study. Columns represent 5 min intervals. Sedation and vital sign parameters assessment were performed every 5 min from 15 min prior to dexmedetomidine administration (T0), to post-dexmedetomidine at 120 min Atipamezole was then administered (Ta)and continued for 30 min. The responsiveness assessment was performed every 15 min pre-dexmedetomidine and post-dexmedetomidine, until after the atipamezole injection.

**Figure 2 animals-12-02787-f002:**
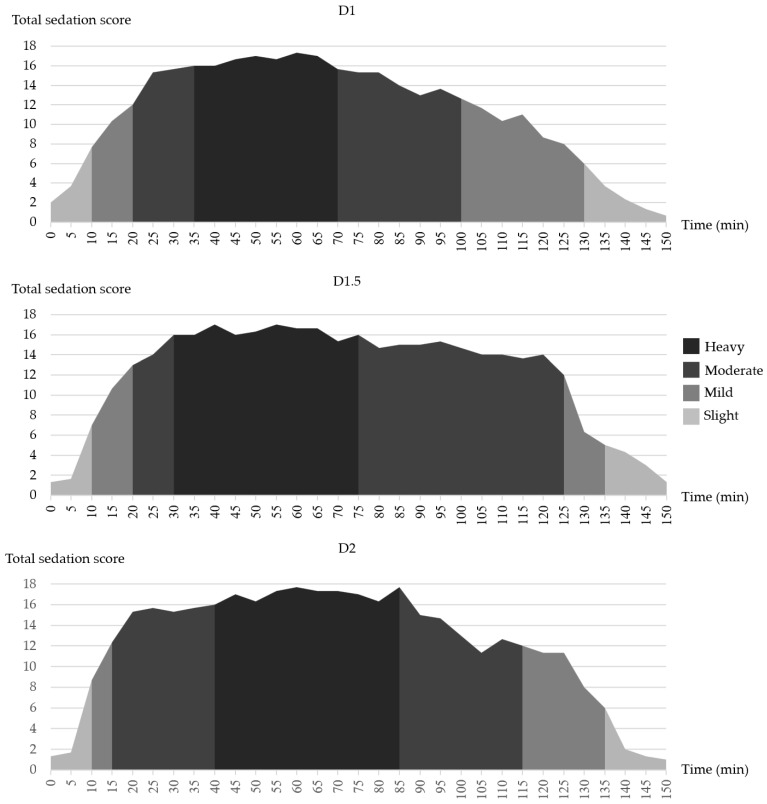
Area charts indicated an average sedation score and sedation level of groups D1, D1.5 and D2 at various time points. The sedation score was divided into four levels, including slight (scores 0–5), mild (scores 6–11), moderate (scores 12–15), and heavy sedation (scores 16–18).

**Figure 3 animals-12-02787-f003:**
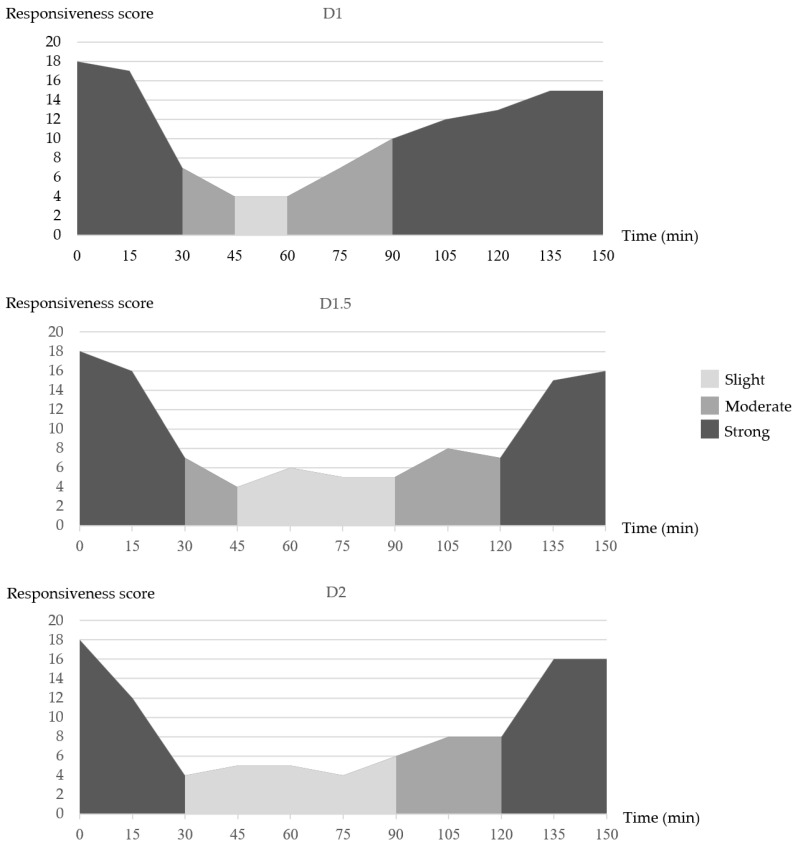
The responsiveness score of elephants in group D1, D1.5, and D2 at different time points. The responsiveness score consisted of three levels of responsiveness classified as slight (scores 0–6), moderate (scores 7–11), and strong responsiveness (scores 12–18).

**Figure 4 animals-12-02787-f004:**
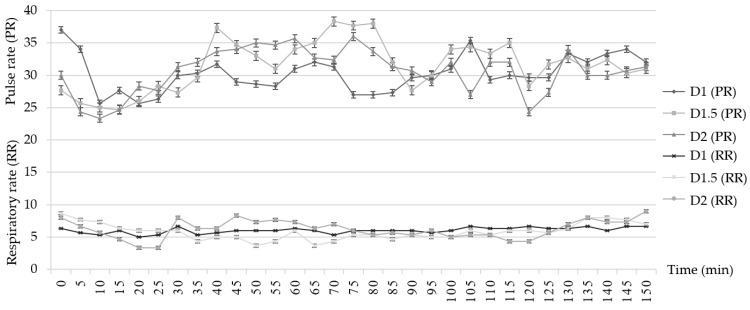
Line charts comparing the average pulse rate and respiratory rate between groups D1, D1.5 and D2 at various time points.

**Table 1 animals-12-02787-t001:** A crossover experimental design for 3 elephants.

Elephant	Week 1	Week 3	Week 5
Elephant 1	D1	D1.5	D2
Elephant 2	D1.5	D2	D1
Elephant 3	D2	D1	D1.5

**Table 2 animals-12-02787-t002:** Sedation score assessment (modified from Neiffer et al., 2005 [2].

Criterions	0–5 Slight Sedation	6–11 Mild	12–15 Moderate	16–18 Heavy
**Awareness of surroundings**	Yes	Moderately aware	Less aware	Not present
**Standing posture and balance**	Moving, rocking back and forth	Stop moving, moderately rocking back and forth	Standstill, less movement, less rocking	Completely motionless
**Ear movement**	Constant flipping (> 10 times/min)	Moderately flipping (5–10 times/min)	Little flipping (< 5 times/min)	Not present
**Tail movement**	Constant tail movement (> 10 times/min)	Moderate tail movement (5–10 times/min)	Little tail movement (< 5 times/min)	Not present
**Trunk movement**	Constant trunk movement	Moderate trunk movement	Little trunk movement, Relaxed, lying on the ground, occasional motion	Relaxed, lying on the ground, motionless
**Genital protruding**	Intact (no prolapse)	Mild prolapse	Protruding from prepuce (incomplete prolapse)	Protruding from prepuce (complete prolapse)

**Table 3 animals-12-02787-t003:** Responsiveness score assessment (modified from Nishimura et al., 2018 [16]).

Response	0–6 Slight	7–11 Moderate	12–18 Strong
**Palpebral reflex**	Intermittent or slow response, sluggish	Moderate reflex	Present and strong
**Eye**	Slightly open, able to lift eyelid, third eyelid intact	Moderately open	Wide open
**Ear tone** **(Gently pull the ears)**	Little response/ flipping ears	Moderate response/ flipping ears	Strong response/ flipping ears
**Jaw tone** **(Gently pull lower lip to open the mouth)**	Slight resistance	Moderate resistance	Strong resistance
**Tail tone** **(Lifting tail to observe resistance)**	Slight resistance	Moderate resistance	Strong resistance
**Pain reflex** **(Pinch forelimb skin with a hemostat forceps)**	Slight resistance	Moderate resistance	Strong resistance

## Data Availability

Not applicable.

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
