# Peer review of "Dexmedetomidine Effectively Sedates Asian Elephants (*Elephas maximus*)"

_animals, 2022, doi:10.3390/ani12202787_

Round 1
Reviewer 1 Report
The largest problem I had reviewing this manuscript was sentence structure and sentence arrangement. There were several sentences that had to be reread multiple times to attempt to understand what the authors were saying. Because of this, I strongly recommend a revision to address the language of the article and perhaps a review by a native English speaker prior to resubmission.
Specific concerns:
1) Sentence structure with run-on sentences or sentences with randomness in their structure that made understanding difficult. There was a lack of clarity and conciseness throughout the manuscript. Rather than list all of the specific sentences, I strongly encourage the authors to specifically look at clarity and conciseness of the manuscript sentence structure. If need be, consider consulting with a native English speaker for assistance in revision.
An example of sentences causing concern: In the abstract it appears that 3 elephants were chosen, and each received a single dose of dexmedetomidine, not that each elephant received 3 different doses with a washout period between doses (lines 33-36). This was explained in the manuscript but is important information lacking from the abstract.
2) Words used in sentences sometimes were inappropriate for the sentence. For example, lines 142-143 states that multiple adverse clinical signs were observed throughout the study, while they were in fact monitored for. There are other instances in the manuscript where words presented may cause a reader to appear to reach a conclusion opposite of what the authors intend.
There are also concerns about the drug dexmedetomidine. This drugs sedation scores after administration to elephants are described, but the possibility of stimulation resulting in animal arousal is not addressed. While the there was an arousability score assigned, did this evaluation mimic the arousal that would occur from performing a medical procedure? Was this seen with the elephants? If not, is this a concern administering this drug as the only drug for sedation?
A second concern about dexmedetomidine is there is a possibility of a ceiling effect seen in some species administered this drug. Was that seen in this study or is it a concern with this study?
Overall, I do think the information in this manuscript has merit and contains important information. The mechanics of the manuscript (i.e., sentence structure/English language) need revision. I encourage the authors to revise this manuscript.
Author Response
Reviewer #1
Comments and Suggestions for Authors
1) Sentence structure with run-on sentences or sentences with randomness in their structure that made understanding difficult. There was a lack of clarity and conciseness throughout the manuscript. Rather than list all of the specific sentences, I strongly encourage the authors to specifically look at clarity and conciseness of the manuscript sentence structure. If need be, consider consulting with a native English speaker for assistance in revision.
Answer: The manuscript was revised for clarity and conciseness by native English speakers.
An example of sentences causing concern: In the abstract it appears that 3 elephants were chosen, and each received a single dose of dexmedetomidine, not that each elephant received 3 different doses with a washout period between doses (lines 33-36). This was explained in the manuscript but is important information lacking from the abstract.
Answer: The clarity of information about experimental design have been added in the abstract. Line 31-34 –
“A crossover design study was performed in 3 Asian elephants. Each elephant was assigned to 1 of 3 treatments groups – 1 (D1), 1.5 (D1.5) or 2 (D2) µg/kg dexmedetomidine (intramuscular injection, IM) with a two-week ‘washout period’ between doses.”
2) Words used in sentences sometimes were inappropriate for the sentence. For example, lines 142-143 states that multiple adverse clinical signs were observed throughout the study, while they were in fact monitored for. There are other instances in the manuscript where words presented may cause a reader to appear to reach a conclusion opposite of what the authors intend.
Answer: Thank you very much for the suggestion. Line 142-143; The sentences were revised. We replaced the word ‘were observed’ with ‘were monitored’ in order to declare that we monitored the clinical signs in elephants throughout the study and 3 days post-study continuously –
“Elephants were monitored for other clinical signs (hypersalivation, shivering, vomiting, nystagmus, and urination) throughout the study and 3-days post-injection.”
There are also concerns about the drug dexmedetomidine. This drugs sedation scores after administration to elephants are described, but the possibility of stimulation resulting in animal arousal is not addressed. While the there was an arousability score assigned, did this evaluation mimic the arousal that would occur from performing a medical procedure? Was this seen with the elephants? If not, is this a concern administering this drug as the only drug for sedation?
Answer: We are much grateful for your notice. During the experiment, we tried to avoid all the stimulants e.g., noise, human voice, movement etc. However, the snoring sound of another elephant under sedation was the important arousal, which stimulated the response of a particular elephant during the experiment. But this did not affect the results, as the snoring occurred for short period (sec) and then ceased. We further added this important item in the discussion section.
(Line 278-282)
“During the experiment, we tried to avoid all stimulants e.g., noise, human voice, movement etc. However, the snoring sound of another elephant under sedation occurred, which stimulated a response of the particular elephant during the experiment. The snoring does not affect the results, as it happened for a second and then disappeared.”
A second concern about dexmedetomidine is there is a possibility of a ceiling effect seen in some species administered this drug. Was that seen in this study or is it a concern with this study?
Answer: Although there is a possibility of a dexmedetomidine ceiling effect, seen in dogs for example (PMID 10747239), at dosages used in this study, we did not observe a ceiling effect. Further studies are needed to investigate the possibility of a ceiling effect in elephants.
Overall, I do think the information in this manuscript has merit and contains important information. The mechanics of the manuscript (i.e., sentence structure/English language) need revision. I encourage the authors to revise this manuscript.
Answer: Thank you very much.

Reviewer 2 Report
The manuscript is well organized and important information is communicated regarding dexmedetomidine sedation in Asian elephants. However, I would recommend the following changes.
Line 42 - change to read: standing sedation
Line 68 - remove use and change to: effect and remove not known to unknown
Line 76 - providing actual ages of the elephant and their weights would be helpful to readers or at least giving the actual ranges in addition to the current information.
line 113 - there are no vertical gray bars in the figure 1. Recommend changing to columns represent 5 min intervals.
Line 129 - The responsiveness table does not end at 18, as indicated in your description, but goes to 23. You need to redo the table either to match your description, or use the 0-5 slight, 6-11 is moderate and 12-23 as strong responsiveness. This same issue occurs on lines 133, 172, 174, and 196.
143 - recommend changing to increase clarity: Elephants were observed for other clinical signs (......)
Line 151 - recommend changing to: P value ≤0.05
Line 202 - suggest adding ±SD.
Line 209 - suggest changing to: ..first study to examine the effects....
Line 287 - suggest removing the work ilk, as it adds nothing to the statement and in a misuse of the term.
Thank you for your efforts to publish this information.
Author Response
Reviewer #2
The manuscript is well organized and important information is communicated regarding dexmedetomidine sedation in Asian elephants. However, I would recommend the following changes.
Line 42 - change to read: standing sedation
Answer: Thank you very much for the suggestion. Line 41 (scrolled) was revised.
Line 68 - remove use and change to: effect and remove not known to unknown
Answer: Thank you very much for the suggestion. Line 66 (scrolled) was revised.
Line 76 - providing actual ages of the elephant and their weights would be helpful to readers or at least giving the actual ranges in addition to the current information.
Answer: Thank you very much for the suggestion. Line 75-77 (scrolled); The information has been added. –
“Three healthy, non-musth male Asian elephants (elephant 1 body weight (BW) 2,870 kg, age 28 years; elephant 2 BW 2,900 kg, age 20 years; elephant 3 BW 3,865 kg, age 27 years, average age, 25.7 ± 5.1 years old, average BW 2800 ± 900.5 kg)…”
Line 113 - there are no vertical gray bars in the figure 1. Recommend changing to columns represent 5 min intervals.
Answer: Thank you very much for the suggestion. Line 116 (scrolled); This has been revised.
Line 129 - The responsiveness table does not end at 18, as indicated in your description, but goes to 23. You need to redo the table either to match your description, or use the 0-5 slight, 6-11 is moderate and 12-23 as strong responsiveness. This same issue occurs on lines 133, 172, 174, and 196.
Answer: L133 (scrolled); The table has been corrected. We adjusted the table score as shown in table 3 with score range 0-18 (from slight to strong response) in order to match with sedation score of more sedation, the less response. The level and score range shown in description is correct, also on line 133, 172, 174, and 196. Table was matched with description.
Table 3. Responsiveness score assessment (modified from Nishimura et al., 2018 [16])
|
Response |
0 – 6 slight |
7 – 11 moderate |
12 – 18 strong |
|
Palpebral reflex |
Intermittent or slow response, sluggish |
Moderate reflex |
Present and strong |
|
Eye |
Slightly open, able to lift eyelid, third eyelid intact |
Moderately open |
Wide open |
|
Ear tone (Gently pull the ears) |
Little response/ flipping ears |
Moderate response/ flipping ears |
Strong response/ flipping ears |
|
Jaw tone (Gently pull lower lip to open the mouth) |
Slight resistance |
Moderate resistance |
Strong resistance |
|
Tail tone (Lifting tail to observe resistance) |
Slight resistance |
Moderate resistance |
Strong resistance |
|
Pain reflex (Pinch forelimb skin with a hemostat forceps) |
Slight resistance |
Moderate resistance |
Strong resistance |
Line 143 - recommend changing to increase clarity: Elephants were observed for other clinical signs (......)
Answer: Thank you very much for the suggestion. Line 142-143; The sentences were revised. We replaced the word ‘were observed’ with ‘were monitored’ in order to declare that we monitored the clinical signs in elephants throughout the study and 3 days post-study continuously –
“Elephants were monitored for other clinical signs (hypersalivation, shivering, vomiting, nystagmus, and urination) throughout the study and 3-days post-injection.”
Line 151 - recommend changing to: P value < 0.05
Answer: Thank you very much for the suggestion. However, the data analysis was edited.
Data are presented as mean ± standard error of the mean (SEM). The differences in means of sedation score and vital sign values among drug doses groups were compared using ANOVA. Since the Latin square design is employed, the ANOVA model corresponds to this design is as follows:
where is observation from animal receiving drug at period while is overall mean. The term denote individual effect of animal ( =1, 2 and 3) and represent effects from period ( =1, 2 and 3), while is the effects from drug doses ( = 1 µg/kg, 1.5 µg/kg and 2 µg/kg). The term is residual and it was assumed that . For multiple comparisons (Post Hoc), the Bonferroni method was used. Additionally, the differences in means among stages relevant to the drug administration stages (1 = before, 2 = during and 3=after) across all doses were compared. The level of significant was set as for all statistical analyses.
Line 202 - suggest adding ±SD.
Answer: Thank you very much for the suggestion. Line 196-197 (scrolled); This has been added.
Line 209 - suggest changing to: ...first study to examine the effects....
Answer: Thank you very much for the suggestion. Line 203 (scrolled); This has been corrected.
Line 287 - suggest removing the work ilk, as it adds nothing to the statement and in a misuse of the term.
Answer: Thank you very much for the suggestion. Line 284-286 (scrolled); This has been revised.

Reviewer 3 Report
This study includes a statistical issue regarding the number of animals investigated. The authors reported that normality and homogeneity of variance were tested, but that result was not shown. Anyway, I doubt the test was meaningful because of a too small number of animals. It was meaningless that the data obtained from only 3 animals were indicated as mean and standard ± error.
I agree with an fact that the determination of the appropriate number of animals is not easy in vivo study, even though a power calculation suggested enough numbers to analyze. On the other hand, in another study with elephants, a larger number of animals could be used. (J Zoo Wildl Med. 2021 Jun;52(2):437-444. doi: 10.1638/2020-0170. PMID: 34130385.) If there were a reasonable reason that the number of elephants investigated should be restricted to such small, the authors must state it in the manuscript.
Furthermore, in the materials and methods section, the description of the used statistical method was not enough or incorrect, which also could not cover the data presented in the manuscript.
However, the purpose of this study is clear, and the protocol seems appropriate except for the number of animals investigated. The result and clinical relevance obtained from this study would be really significant, as long as the statistical issues are resolved.
Author Response
Reviewer #3
This study includes a statistical issue regarding the number of animals investigated. The authors reported that 0.05 were tested, but that result was not shown. Anyway, I doubt the test was meaningful because of a too small number of animals. It was meaningless that the data obtained from only 3 animals were indicated as mean and standard ± error.
Answer: Thank you very much for the important comments. We were limited in the number of animals used in the experiment leading to such small data for analysis and low power of calculation. However, parametric statistical analysis is an essential tool for reporting the results. We decided to report the results as mean and standard ± error because the data we obtained were not much different and there was no difference in summary.
I agree with a fact that the determination of the appropriate number of animals is not easy in vivo study, even though a power calculation suggested enough numbers to analyze. On the other hand, in another study with elephants, a larger number of animals could be used. (J Zoo Wildl Med. 2021 Jun;52(2):437-444. doi: 10.1638/2020-0170. PMID: 34130385.) If there were a reasonable reason that the number of elephants investigated should be restricted to such small, the authors must state it in the manuscript.
Answer: The limitation was additionally stated in the manuscript in Line 300-303. –
“the number of elephants was limited due to the allowance of elephants available to use in the experimental study. The advantage in this study was to compare sedative effects from three different doses used on the same elephant”
Furthermore, in the materials and methods section, the description of the used statistical method was not enough or incorrect, which also could not cover the data presented in the manuscript.
Answer: Thank you very much for the important comments. We have revised the description of statistical analyses by adding more details. Accordingly, the statistical model was shown and each term of presented in the statistical model was defined. We also addressed the Post Hoc as well as the level of significant. Thus the data analysis was rewritten as follows (Line 145-159):
Data are presented as mean ± standard error of the mean (SEM). The differences in means of sedation score and vital sign values among drug doses groups were compared using ANOVA. Since the Latin square design is employed, the ANOVA model corresponds to this design is as follows:
where is observation from animal receiving drug at period while is overall mean. The term denote individual effect of animal ( =1, 2 and 3) and represent effects from period ( =1, 2 and 3), while is the effects from drug doses ( = 1 µg/kg, 1.5 µg/kg and 2 µg/kg). The term is residual and it was assumed that . For multiple comparisons (Post Hoc), the Bonferroni method was used. Additionally, the differences in means among stages relevant to the drug administration stages (1 = before, 2 = during and 3=after) across all doses were compared. The level of significant was set as for all statistical analyses.
However, the purpose of this study is clear, and the protocol seems appropriate except for the number of animals investigated. The result and clinical relevance obtained from this study would be really significant, as long as the statistical issues are resolved.
Answer: Thank you very much.

Round 2
Reviewer 1 Report
The authors have adequately addressed my concerns with the previous review of this manuscript. The manuscript will need a final review for the occasional grammatical/typographic error, but overall this is much improved. Thank you for the opportunity to review this important information that is needed for the captive management of Asian elephants.
Author Response
Answer: Thank you very much for your valuable time to review this manuscript. We repeated review the manuscript as your suggestion. The language was checked by two supporting staff from Stanford University, as well as from Dr. Patrick Sharp from University of California, Merced, USA, one of the co-authors. All of them are English native speakers. Thank you very much for your valuable time to review this manuscript.

Reviewer 3 Report
I do not accept the authors' explanation regarding the number of animals and the presentation of data. And I do not know why parametric statistics is essential or suitable. The authors could also show the result as each elephant data but not the mean value.
In my recommendation, when the number of elephants could not be increased, the authors should present the data for each individual value and analyze it with nonparametric statistical methods.
Author Response
Answer: Thank you very much for your useful comments. We acknowledged that you are right that the non-parametric methods should be used in case of small sample size. Nevertheless, this study was conducted using Latin Square Design (LSD), thus the most appropriate method is ANOVA model accounted for the LSD as recommended by well know text books, for instance please see (Dean & Voss, 1999; Montgomery, 2017). By using 3x3 LSD, the number of treatment (n=3) is equal to the number of animals used in the study (n=3) and thus the experiment was conducted 3 periods. Each animal will receive a different treatment at each period. The transition from one period to the nest period is done by including wash-out period.
We followed the statistical concept addressed in several text books that “Experimental Design dictates the analysis”. Thus, we analyzed the data by just using ANOVA but ANOVA for LSD design. We addressed the statistical model rather than just mention about ANOVAS to make a clear statement on the analysis. Not only treatment effects were tested but also effects due to individual and period were retained in the model (Dean & Voss, 1999; Montgomery, 2017). Additionally, numerous studied performed the similar approach as we did in this study. Please see (Cherdthong & Wanapat, 2013; Desevaux et al., 2017; Johnson et al., 2007; Phesatcha et al., 2022; Valente et al., 2021; Wagner et al., 2020) for an example. Additionally, other reviewers did not mention about the using of non-parametric. Thus, we acknowledge their assessment that the change in statistical analysis is not required. Finally, we would be greatly grateful if you would examine our efforts to do the analysis using models that are most appropriate to the design, as indicated in text books and generally utilized by numerous research groups.
Cherdthong, A., & Wanapat, M. (2013, 2013/12/01). Rumen microbes and microbial protein synthesis in Thai native beef cattle fed with feed blocks supplemented with a urea–calcium sulphate mixture. Archives of Animal Nutrition, 67(6), 448-460. https://doi.org/10.1080/1745039X.2013.857080
Dean, A., & Voss, D. (1999). Design and analysis of experiments. Springer.
Desevaux, C., Marotte-Weyn, A. A., Champeroux, P., & King, J. N. (2017, Dec). Evaluation of cardiovascular effects of intravenous robenacoxib in dogs. J Vet Pharmacol Ther, 40(6), e62-e64. https://doi.org/10.1111/jvp.12411
Johnson, J. A., Robertson, S. A., & Pypendop, B. H. (2007, 01 Jul. 2007). Antinociceptive effects of butorphanol, buprenorphine, or both, administered intramuscularly in cats. American Journal of Veterinary Research, 68(7), 699-703. https://doi.org/10.2460/ajvr.68.7.699
Montgomery, D. C. (2017). Design and analysis of experiments. John wiley & sons.
Phesatcha, K., Phesatcha, B., Wanapat, M., & Cherdthong, A. (2022). The Effect of Yeast and Roughage Concentrate Ratio on Ruminal pH and Protozoal Population in Thai Native Beef Cattle. Animals, 12(1), 53. https://www.mdpi.com/2076-2615/12/1/53
Valente, E. E. L., Damasceno, M. L., Klotz, J. L., & Harmon, D. L. (2021). Residual effects of abomasal 5-hydroxytryptophan administration on serotonin metabolism in cattle. Domestic Animal Endocrinology, 76, 106627. https://doi.org/https://doi.org/10.1016/j.domaniend.2021.106627
Wagner, B. K., Relling, A. E., Kieffer, J. D., Moraes, L. E., & Parker, A. J. (2020). Short communication: pharmacokinetics of oxytocin administered intranasally to beef cattle. Domestic Animal Endocrinology, 71, 106387. https://doi.org/https://doi.org/10.1016/j.domaniend.2019.106387

Round 3
Reviewer 3 Report
Thank you very much for your elaborate explanation regarding the statistical issue. I would like to apologize for my shortage of statistical knowledge.
I have accepted your claim because of the sufficient evidence you indicated. I am happy not to disturb the publication of your work such informative.